# Weak Cross-Lineage Neutralization by Anti SARS-CoV-2 Spike Antibodies after Natural Infection or Vaccination Is Rescued by Repeated Immunological Stimulation

**DOI:** 10.3390/vaccines9101124

**Published:** 2021-10-02

**Authors:** Sara Caucci, Benedetta Corvaro, Sofia Maria Luigia Tiano, Anna Valenza, Roberta Longo, Katia Marinelli, Monica Lucia Ferreri, Patrik Spiridigliozzi, Giovanna Salvoni, Patrizia Bagnarelli, Stefano Menzo

**Affiliations:** 1Department of Biomedical Sciences and Public Health, Università Politecnica delle Marche, 60126 Ancona, Italy; s.caucci@univpm.it (S.C.); benedettacorvaro@virgilio.it (B.C.); sofiamarialuigia.tiano@gmail.com (S.M.L.T.); p.bagnarelli@univpm.it (P.B.); 2Virology Laboratory, Azienda Ospedaliera Ospedali Riuniti di Ancona, 60126 Ancona, Italy; anna.valenza@ospedaliriuniti.marche.it (A.V.); katia.marinelli@ospedaliriuniti.marche.it (K.M.); monicalucia.ferreri@ospedaliriuniti.marche.it (M.L.F.); patrik.spiridigliozzi@ospedaliriuniti.marche.it (P.S.); giovanna.salvoni@ospedaliriuniti.marche.it (G.S.)

**Keywords:** COVID, SARS, neutralizing antibodies, BAU, anamnestic response, vaccine, vaccine dose

## Abstract

After over one year of evolution, through billions of infections in humans, SARS-CoV-2 has evolved into a score of slightly divergent lineages. A few different amino acids in the spike proteins of these lineages can hamper both natural immunity against reinfection, and vaccine efficacy. In this study, the in vitro neutralizing potency of sera from convalescent COVID-19 patients and vaccinated subjects was analyzed against six different SARS-CoV-2 lineages, including the latest B.1.617.2 (or Delta variant), in order to assess the cross-neutralization by anti-spike antibodies. After both single dose vaccination, or natural infection, the neutralizing activity was low and fully effective only against the original lineage, while a double dose or a single dose of vaccine, even one year after natural infection, boosted the cross-neutralizing activity against different lineages. Neither binding, nor the neutralizing activity of sera after vaccination, could predict vaccine failure, underlining the need for additional immunological markers. This study points at the importance of the anamnestic response and repeated vaccine stimulations to elicit a reasonable cross-lineage neutralizing antibody response.

## 1. Introduction

Since the spillover to humans of SARS-CoV-2, and its pandemic diffusion, COVID-19 has wreaked havoc in most countries, taking a particular toll on human lives in industrialized countries, where a large share of the population exceeds 65 years of age. To date, no effective treatment strategy has been identified that is capable of neutralizing the pathogenic potential of the infection in high-risk populations on a large scale. The only feasible strategy to curb mortality and wean the world from restrictive measures will be the efficacy of vaccination campaigns. In its spread across the continents, through billions of infections (and infinitely more replicative cycles), SARS-CoV-2 inevitably evolved in a few clades and a score of lineages, despite the proofreading replication and low overall mutation rate (compared to other RNA viruses) common to all coronaviruses. Some of these lineages have proven to be more contagious, and also capable of overcoming (by reinfecting subjects who had already recovered from the infection) the partial herd immunity reached in many countries where restrictive measures could not be effectively implemented, such as South Africa, Brazil, and India. As none of these lineages were circulating during the registration trials for the approved vaccines [1,2,3,4], the impact of the mutations in the spike gene of these lineages on the efficacy of these vaccines is still a matter of scientific debate [1]. Among the mutations observed in the spike gene, a few appeared consistently in unrelated lineages in a process of converging evolution, such as E484K/R, N501Y, K417T, and others, suggesting a specific selective pressure on these residues that are located at the spike interface with the ACE2 (Angiotensin Converting Enzyme 2) receptor. The location of such mutations implies a potential deleterious impact on the binding, and on the efficacy, of antibodies with neutralizing capability. A growing score of data obtained from in vitro neutralization studies has already been published [2,3,4,5,6], albeit some were obtained with a pseudotyped virus rather than with real viral isolates, demonstrating variable losses of the in vitro neutralizing potency of sera from convalescents and vaccinees. However, data from in vivo studies are still largely insufficient, even for the more established lineages. A few recent studies performed in countries where the prevalence of vaccination is high, and the B.1.1.7 lineage is circulating, suggest, for example, that the vaccine has effectively protected the population from severe disease as well as, to some extent, from infection [7,8,9],while very recently a study analyzed the impact of the B.1.617.2 lineage in the UK [10]. However, these results are limited by the extremely short observation times, by the use of methods for establishing efficacy that are necessarily less reliable than those employed in clinical trials, and are yet to be confirmed for other populations, other vaccines, and other lineages. Another issue that will be crucial for the future management of vaccination campaigns is to what extent, and by which simple means, we can predict vaccine efficacy. The identification of a predictive marker will allow for design vaccination recall policies for sustaining effective immunity in single individuals, and herd immunity in the population. To address these issues, we investigated the in vitro neutralizing power and binding activity of sera from both vaccinated individuals (some of whom experienced vaccine failure), and those who have recovered from natural infection, measured against viral isolates of some of the recently emerged lineages.

## 2. Materials and Methods

### 2.1. Virus Stocks

Different lineages of SARS-CoV-2 were obtained from RT-PCR positive nasopharyngeal swabs collected at Ospedali Riuniti, Ancona, Italy. The isolation of SARS-CoV-2 was performed using Vero E6 cells (ATCC n° CRL-1586), as described by [11]. Supernatants of the infected cells were harvested at 80% CPE (2–4 days after infection), centrifuged at 3000 rpm for 10 min, filtered using a 0.2 µm filter, divided in aliquots, and stored at −80 °C. The complete genome’s ore spike sequences, and the viral stocks, were sequenced on different NGS platforms (Ion Torrent S5, Illumina Myseq, MinIon). Five lineages, according to the Pangolin lineage nomenclature [12], were used for the present study: B.1 (EPI_ISL_417491), B.1.1.7 (EPI_ISL_778869), P.1 (EPI_ISL_1118260), B.1.351 (EPI_ISL_1118258), B.1.526 (EPI_ISL_1321993), and B.1.617.2 (EPI_ISL_2975994).

### 2.2. Cells and Cell Cultures

Vero E6 cells were cultured in Dulbecco’s modified Eagle’s medium (DMEM), supplemented with 10% fetal calf serum (FCS) and antibiotic/antimycotic solution (growth medium). The cells were split weekly, and cultures were incubated at 37 °C under a 5% CO^2^ atmosphere.

To determine the fifty-percent tissue culture infectious dose (TCID_50_) of the viral stocks, cells (seeded 24 h earlier at 2.6 × 10^4^/well in 96-well microplate) were incubated with 50 μL of ten-fold serial dilutions in DMEM, ranging from 10^−1^ to 10^−8^ of viral stocks, for 2 h. Following incubation, the cells were washed twice, and growth medium was added. After 72 h of incubation, the Reed-Muench method was used to calculate the TCID_50_/mL for each viral stock.

### 2.3. Kinetics of SARS-CoV-2 Replication in Vero E6 Cells

Cells were seeded and infected after 24 h with 100 TCID_50_ of viral stocks, 50 µL per well, for 2 h after removal of the growth medium. The inoculum was removed, the cells were washed twice, and 100 µL of fresh medium was added. The supernatants were collected after 6, 12, 24, 48, and 72 h for the quantitative RT-PCR.

### 2.4. Quantitative Real Time RT-PCR

The QIAsymphony automated platform (QIAGEN, Hilden, Germany) was used to extract viral RNA using the Kit QIAsymphony DSP Virus/Pathogen Midi kit according to the manufacturer’s instructions. Quantitative real-time PCR was performed on an Applied BiosystemsTM7500 Fast Dx Real-Time PCR Instrument (Thermo Fisher Scientific, Waltham, MA, USA) using a commercial set of primers and probes (Integrated DNA Technologies, cat# 10006770), based on the protocol issued by the CDC [13]. Quantitation was achieved with a calibration curve based on 10-fold dilutions (10^5^ to 10^2^ cps/rct) of the WHO International Standard for SARS-CoV-2 RNA (cat# 20/146), purchased from the National Institute for Biological Standards and Control (NIBSC), included in each session.

### 2.5. SARS-CoV-2 Neutralization Assay

Vero E6 cells were plated in 96-well plates, at 2.6 × 10^4^ cells per well, 24 h before infection.

Sera were inactivated at 56 °C for 30 min and were serially two-fold diluted from 1:10 to 1:640 in DMEM 10% FCS (?). Viral stocks used at 100 TCID_50_ in 50 µL were incubated (1:1) with serum dilutions in triplicate for 1 h at 37 °C. The 100 µL antibody-virus mixtures were subsequently added to the Vero E6 cells after medium removal. After 72 h of incubation, the infected wells were counted, based on the typical cytopathic effects, and recorded for each serum dilution. Titers were calculated, by interpolating in an exponential curve the frequency of replicate wells where a CPE was observed (up to 100%), to establish the virtual dilution inhibiting infection in 50% of wells.

### 2.6. Antibody Binding Assay

A commercial automated chemiluminescent assay (SARS-CoV-2 IgG II, Abbott Abbott Science Park, Chicago, IL, USA) was used to quantify the antibody binding activity, expressed in binding antibody units (BAU/mL).

### 2.7. Human Sera

Convalescent sera were collected in the context of the national Tsunami trial (transfusion of convalescent plasma) for the treatment of severe pneumonia due to SARS-CoV-2), a randomized controlled multicentric study evaluating the safety and efficacy of the use of convalescent plasma for the cure of COVID-19.

Sera from vaccinees were collected in the context of the multicentric study, “Monitoraggio della vaccinazione anti-SARS-CoV-2 in una coorte di operatori sanitari: efficacia sul campo e risposta immunitaria”, on the efficacy of vaccines in healthcare workers, promoted by the National Institute for Infectious Diseases, “Lazzaro Spallanzani”.

Both studies were approved by the ethics committee of the National Institute for Infectious Diseases, “Lazzaro Spallanzani”, and by the ethics committee of region Marche (CERM). Informed consent was obtained from all subjects involved.

Sera from healthcare workers receiving the BNT162b2 (January–March 2021) were obtained from blood drawn 21 days after the 1st dose, and/or 15 (+/−1) days after the 2° dose, and were divided into the following groups: 50 with no previous SARS-CoV-2 infection (negative for anti-nucleocapsid), 15 with a PCR-confirmed SARS-CoV-2 infection between February and December 2020, and 12 who experienced vaccine failure with PCR-confirmed SARS-CoV-2 infection in the interval between the 3° and the 13° week after the second dose. An additional 20 sera from subjects vaccinated with the first dose of the AZD1222 vaccine (of which 10 were vaccinated after natural infection) were obtained three weeks after inoculation. 33 convalescent sera were obtained from blood donors (region Marche, Italy), between June and September 2020, who experienced PCR-confirmed SARS-CoV-2 infection during the period between March and May 2020. Additional convalescent sera were obtained from donors infected with the B.1.1.7 (*n* = 13), P.1 (*n* = 1), and B.1.351 (*n* = 1) variants during the period from January–March 2021. The age and gender, and any relevant COVID-19 history pertaining to the subjects (divided into two groups), are included in this study and listed in Appendix A

### 2.8. Statistical Analysis

Statistical analyses were performed using GraphPad Prism (Graphpad Software, San Diego, CA, USA). Nonparametric tests were applied to compare groups, either the Wilcoxon matched pairs signed rank test, where appropriate, or the Mann-Whitney rank test. The statistical significance was defined as *p* < 0.05.

## 3. Results

### 3.1. Cross-Lineage Neutralizing Activity of Vaccine- and Infection-Induced Antibodies

SARS-CoV-2 of different lineages (B.1, B.1.1.7, B.1.351, P1, B.1.526, and B.1.617.2) was isolated from infected patients referred to the University Hospital of Ancona, Italy. The primary isolates were subsequently synchronized in parallel cultures to evaluate the phenotypic aspects of the infected cultures. In these conditions, the different lineages displayed different replication kinetics (Appendix A). Noticeable differences in the cytopathic effects between the lineages were also apparent. In particular, B.1.1.7 and B.1.617.2 not only displayed a delayed replication kinetics but were also more markedly syncytiogenic (Appendix A). To obtain the best possible standardization throughout this study, the neutralizing power of sera was evaluated against isolates grown and titered in the same experiment and tested in parallel for all lineages.

The neutralizing efficacy against the different lineages of antibodies elicited by ancestral SARS-CoV-2 spike proteins, both after vaccination and natural infection, was analyzed. A total of 50 sera from subjects (self-reportedly not previously infected by SARS-CoV-2 and seronegative for anti-nucleocapsid), vaccinated with the Comirnaty (BNT162b2) vaccine (14 days after the second dose), were tested on the five mentioned lineages (Figure 1a), and 33 sera from patients naturally infected before the introduction of the B.1.1.7 lineage in Italy, therefore by the B.1 and B.1.177 lineages (median 110 days, iqr 93–143 days from infection), were tested against four lineages (Figure 1b). A clear decrease (compared to the ancestral strain) in the neutralizing activity of the sera from vaccines could be observed against all “variant” lineages (*p* < 0.0001): B.1.1.7 (3.2-fold), P.1 (3.5-fold), and B.1.526 (2.9-fold). A more evident loss was observed against B.1.617.2 (8.3-fold) and was even more evident against B.1.351 (35-fold), against which most sera from both populations lost their efficacy. Convalescent sera displayed a comparable decrease in neutralizing activity. To cross-check whether the experimental design was adequate for identifying lineage-specific antibody responses, a few sera from patients naturally infected with different lineages were also tested against the same array of lineages (Figure 1c). Indeed, 13 sera from patients infected with the B.1.1.7 lineage (median 31 day, iqr 19–44 days from infection) displayed a significantly higher neutralizing power against that specific lineage, while two sera from P1 and B.1.351 infection were also mostly active against their respective lineages. Notably, the serum from B.1.351 infection was remarkably cross-reactive with all other lineages, which suggests that the spike protein from this lineage might be a better target for eliciting broadly neutralizing antibodies.

In order to investigate whether in vitro neutralizing activity or binding activity could represent the predictive markers of vaccine failures, the results of the sera (obtained two weeks after the second dose, and before the infection) from a small cohort of 12 subjects who experienced COVID-19 after the second dose (median 39 days, iqr 32–45 days) of the BNT162b2 vaccine were compared to those from the general population of vaccinees previously described. The results (Figure 1d) show that, despite a slight (and not statistically significant) reduction of the neutralizing titer of their sera against the original lineage, the neutralizing activity of their sera was not statistically different from that of the whole population of vaccinees for the B.1.1.7 lineage (by which they were infected), and neither was their binding activity (Figure 2).

### 3.2. Anamnestic Response

In order to evaluate the anamnestic response and immunological memory in individuals who had already experienced PCR-confirmed SARS-CoV-2 infection, the sera from fifteen subjects and three subjects, vaccinated with two doses of the BNT162b2 and the AZD1222 vaccines, respectively, after a variable time from the original infection (median 97 days, iqr 81–319), were analyzed. The results (Figure 3a) show that, in this population, the neutralizing titers were all significantly higher (*p* < 0.001) compared to those in the noninfected vaccines. Moreover, the titer improvement against a highly refractory lineage, such as B.1.351 (median 149 vs. 10: 15-fold), was superior to the improvement against other variants. Interestingly, this phenomenon could not be observed by measuring binding activity (Figure 2). The anamnestic response in these subjects appeared stable in time, as it was even more intense (albeit not statistically significant) when natural infection occurred one year earlier compared with six months earlier (Figure 3b). To gain more insight into the anamnestic neutralizing antibody response, we had the opportunity to analyze subjects who were infected between March and December 2020 and vaccinated with the first dose of the BNT162b2 vaccine (*n* = 3), or with a first dose of the AZD1222 vaccine (*n* = 10). All responded with both binding and neutralizing titers comparable to, or exceeding, those obtained after a double dose in naïve subjects, in terms of the breadth of response against the different lineages (Figure 3c).

Again, there was no difference between the five who experienced infection one year before vaccination (March 2020), and the five who experienced it 4–6 months before (October–December 2020). Remarkably (although the limited number of observations do not allow statistical analysis), and in contrast to what was observed in naïve vaccinated subjects (Figure 4a,b), in subjects vaccinated after natural infection, neutralizing titers did not improve after the second dose with either vaccine (Figure 4c,d). Considering that a single dose of either vaccine is mostly ineffective against the tested variant lineages, and that, in naïve patients vaccinated with the AZD1222 vaccine (first or second dose), neutralizing titers are usually much lower than in those vaccinated with the respective dose of the BNT162b2 vaccine, this result further underlines the importance of a solid immunological priming associated with a repeated immunological stimulus (even a modest one), and memory B cells, in boosting this kind of response.

## 4. Discussion

The availability of effective vaccines against COVID-19 will be the solution for gradually alleviating the effects of this devastating pandemic. However, their impact on the population of single countries and globally will depend, not only on the availability of the vaccines and on their administration to the populations, but also on their efficacy against the virus currently spreading in these populations. In more than one year of pandemic, the virus has started a significant evolution in a radiation of lineages throughout the continents, with a growing degree of divergence from the original strain from which all current vaccines were derived. Consequently, a significant reduction in vaccine effectiveness is to be expected. Although the preliminary data on real life populations induce optimism [14,15,16,17,18,19,20], the observation times for these analyses have been extremely short, and their real effectiveness for reasonable timelines are still lacking. Institutional data from Israel suggest that protection against infection of the Delta variant wanes rapidly. As the ongoing vaccination campaigns involve larger, but still partial, shares of the world’s populations (including large shares of natural immunity in some parts of the world), the selective pressure on the virus will grow and may uncover yet more diverging lineages with unpredictable consequences. In this study, we demonstrate, using primary SARS-CoV-2 isolates, that even a limited number of mutations in some lineages (as in the case of B.1.351) can nearly abolish in vitro neutralizing activity, adding to similar knowledge published by other studies [2,3,4,5,21,22]. Despite the limitation of the small number of subjects analyzed, the data from this study strengthens and extends the observation [23] that a single dose of vaccine may confer insufficient protection against some lineages, suggesting that systematically delaying the second dose poses some risks at a population level. On the other hand, the study underscores the beneficial effects of repeated immune stimulations already observed by others [24], as is the case of COVID-19 patients who subsequently produced high-level neutralizing antibodies after an otherwise weakly effective (at least in vitro) vaccination, such as a single dose of the AZD1222 vaccine. Moreover, data from this study demonstrate that the elicited anamnestic response, in terms of neutralizing titers, can be equally as intense after one year than after six months after natural infection, providing a functional demonstration of the persistent presence of long-lived plasma cells and memory B cells [25]. The fact that simple binding activity does not parallel the superior neutralizing power observed in anamnestic responses underlines, also, the role of antibody affinity maturation as a key factor to boost vaccine efficacy. In addition, the study shows that subjects who were infected by SARS-CoV-2 before vaccination do not need two doses of vaccine, even after one year from the infection. This is encouraging for planning reasonable recall schedules for the vaccination campaigns in the future. In the next few years, one of the key points for a rational planning of the vaccination campaigns will be the availability of a reliable and simple marker of immunization efficacy. However, in the present study, neither antibody binding activity (as performed by commercially available chemiluminescence assays), nor in vitro neutralizing activity, could predict insufficient protection and vaccine failure on an individual basis. In the quest for better markers, future studies should concentrate on other aspects of the immune response and, in particular, on cell-mediated responses.

## Figures and Tables

**Figure 1 vaccines-09-01124-f001:**
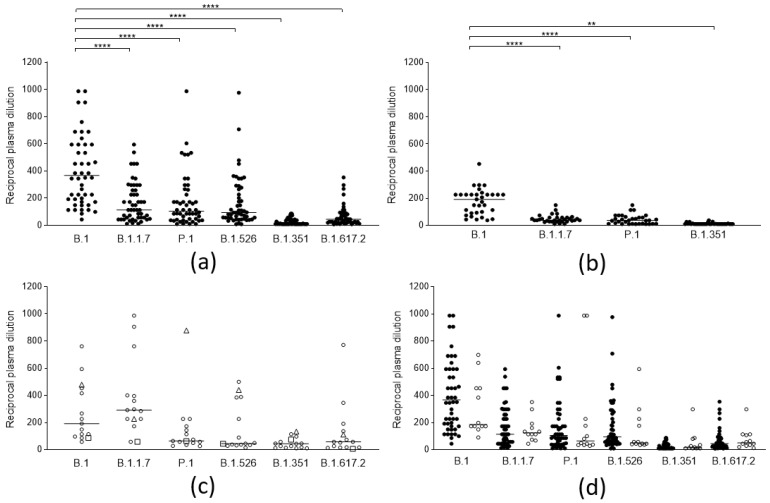
Neutralization activity of SARS-CoV-2 anti-spike antibodies from different groups of subjects against selected viral lineages. (**a**) Neutralization titers of vaccinees (*n* = 50) 2 weeks after the second dose of the BNT162b2 vaccine against 6 viral lineages. Three data points are outside the axis limits. Statistical difference was assessed by the Wilcoxon matched pairs signed rank test. (**b**) Neutralization titers of blood donors naturally infected by the B.1 lineage (*n* = 33) against 4 viral lineages. Statistical difference was assessed by the Wilcoxon matched pairs signed rank test. (**c**) Neutralization titers of blood donors naturally infected by the B.1.1.7 (white circles, *n* = 13), by P.1 (white triangles, *n* = 1), and B.1.351(white squares, *n* = 1) against 6 viral lineages. (**d**) Neutralization titers of subjects who became infected after a second dose of the BNT162b2 vaccine (vaccine failure, white circles, *n* = 12) compared to those from the general population of vaccinees (black circles, *n* = 50). Reductions in neutralizing titers were not statistically significant. Three data points are outside the axis limits. Statistical difference was assessed by the Mann-Whitney rank test. Neutralization titers <20 and >1280 were plotted, respectively, as 10 and 2560. Statistical significance: ns = *p* > 0.05, ** = *p* < 0.01, **** = *p* < 0.0001. Horizontal lines represent medians.

**Figure 2 vaccines-09-01124-f002:**
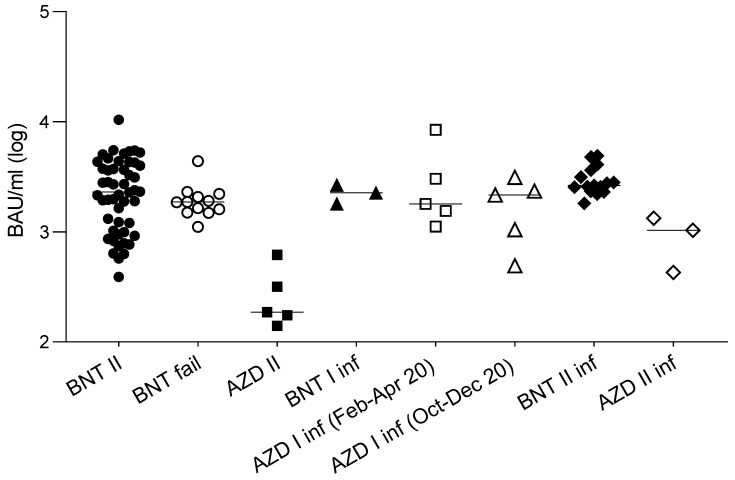
Antibody binding activity of sera from different groups of subjects. Antibody binding activity of: 50 naïve subjects vaccinated with a double dose of the BNT162b2 vaccine (BNT II: black circles); 12 subjects who became infected after a second dose of the BNT162b2 vaccine (BNT fail: white circles); 5 naïve subjects vaccinated with a double dose of AZD1222 (AZD II: black squares); 3 subjects vaccinated with a single dose of the BNT162b2 after natural infection (BNT I inf; black triangles); subjects vaccinated with a single dose of the AZD1222 vaccine after natural infection contracted between February and April 2020 (AZD I inf (Feb-Apr 20): white squares, *n* = 5), and between October to December 2020 (AZD I inf (Oct-Dec 20): white triangles, *n* = 5); subjects vaccinated with a double dose of the BNT162b2 vaccine after natural infection (BNT II inf: black diamonds *n* = 15), and the AZD1222 vaccine after natural infection (AZD II inf: white diamonds, *n* = 3). Values are expressed in binding antibody units (BAU/mL) according to the WHO standard. Horizontal lines represent medians.

**Figure 3 vaccines-09-01124-f003:**
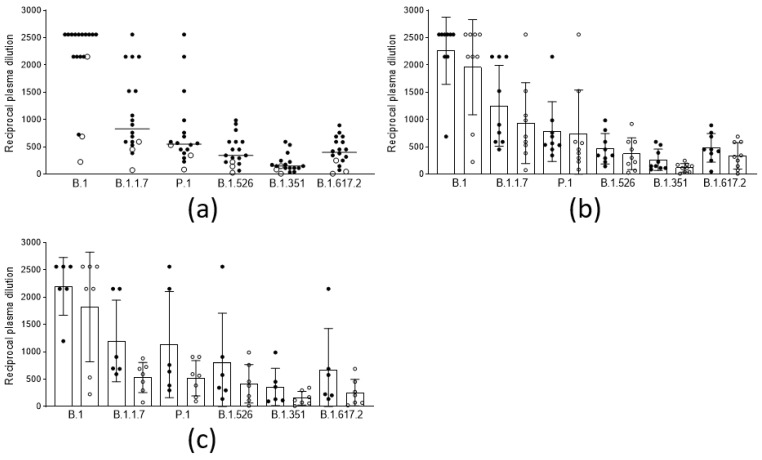
Neutralization titers of sera from vaccinees with anamnestic response and immunological memory who had already experienced PCR-confirmed SARS-CoV-2 infection against different viral lineages. (**a**) Neutralization titers of subjects vaccinated with 2 doses of the BNT162b2 (black circles, *n* = 15), and the AZD1222 (white circles, *n* = 3), vaccines against 6 viral lineages of SARS-CoV-2. Horizontal lines represent medians. (**b**) Neutralization titers against 6 viral lineages of 9 subjects infected between February and April 2020 (black circles) compared to those of 9 subjects infected between October and December 2020 (white circles), both vaccinated with 2 doses of vaccine. Columns represent means, and error bars represent the standard deviation. Reductions of neutralizing titers were not statistically significant. (**c**) Neutralization titers against 6 viral lineages of 6 subjects infected between February and April 2020 (black circles), compared to 7 subjects infected between October-December 2020 (white circles), both vaccinated with 1 dose of vaccine. Columns represent means, and error bars represent the standard deviation. Reductions of neutralizing titers were not statistically significant. Statistical difference was assessed by the Mann-Whitney rank test. Neutralization titers <20 and >1280 were plotted as 10 and 2560, respectively.

**Figure 4 vaccines-09-01124-f004:**
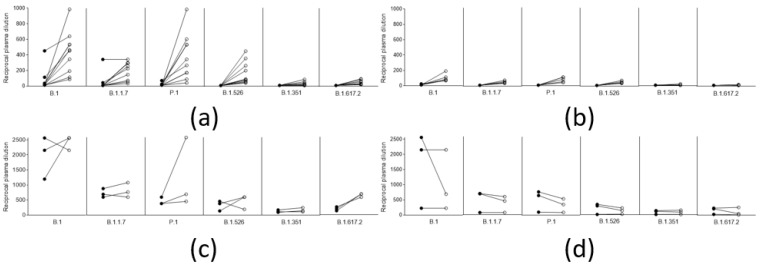
Trend of neutralizing titers against different viral lineages in naive vaccinees and in vaccinees with previous SARS-CoV-2 after a single and a double dose of the respective vaccines. (**a**) Neutralization titers of 10 naïve subjects vaccinated with a single (black circles), and a double dose (white circles), of the BNT162b2 vaccine (**b**) Neutralization titers of 5 naïve subjects vaccinated with a single (black circles), and a double dose (white circles), of the AZD1222 vaccine (**c**) Neutralization titers of 3 subjects vaccinated with a single (black circles), and a double dose (white circles), with the BNT162b2 vaccine after natural infection (**d**) Neutralization titers of 3 subjects vaccinated with a single (black circles), and a double dose (white circles), of the AZD1222 vaccine after natural infection.

## Data Availability

Neutralization data from all subjects have been deposited at Mendeley Data and is publicly available since the 10th of August 2021 (DOI: 10.17632w84245x4cx.1).

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
