# Peer review of "Weak Cross-Lineage Neutralization by Anti SARS-CoV-2 Spike Antibodies after Natural Infection or Vaccination Is Rescued by Repeated Immunological Stimulation"

_vaccines, 2021, doi:10.3390/vaccines9101124_

Round 1

Reviewer 1 Report

The authors screen immune sera from patients both vaccinated for SARS-CoV-2 (BNT162b2 and AZD1222) and convalescent from infection for neutralization activity against the parent/vaccine matched virus and some of the past and present SARS-CoV-2 variants of concern. The results are generally as expected: greater neut for matched virus, and lower coverage for some of the variants, the South African strain in particular. The authors also report some secondary immune responses following vaccination after virus infection.

Major –

  • The pattern of neutralization is expected based on the published literature.
  • It is hard to discern any pattern following secondary exposure through vaccination. There appears to be boosting of memory in some groups, less so in others. Boosting of neutralizing responses in infected individuals has of course been reported in depth.
  • The figure captions are not acceptable:

-figure 1 it  is not clear what (a) through (d) are, and the significance of white vs black, circles vs triangles means. For this reason alone the paper cannot be published in its present form.

-figure 2 caption not acceptable – does not describe data

-Figure 3 and 4 Figure caption is again not acceptable, does not describe the data. Also no stats?

Author Response

Reviewer 1 underlines the lack of originality of this study, and we are aware that he/she is in part right, as of course many other studies have investigated various aspects of humoral response after natural infection and vaccination, this being one of the hot topics of the moment. In fact, the most relevant of these studies have been cited in the bibliography. However, we believe that this study contains some interesting aspects of novelty that deserve attention and will add new knowledge to this fundamental topic.

  • For the first time, neutralizing titers of selected sera were extensively compared across many different lineages (real replicative virus, not surrogate constructs) in the same experiment. This allows to evaluate the quantitative aspects of this extended comparison and provides a scientific interpretation key for their epidemic potential relatively to populations either receiving vaccination or having been extensively hit by previous variants.
  • For the first time, anamnestic response was compared against many different lineages , compared after the first and the second dose of vaccine and between different timings of vaccination from the infection. The notion that anamnestic response after natural infection is relatively stable and not only quantitatively more intense but also qualitatively more proficient against the most refractory variant is completely novel (a comment was added in the results section of the manuscript to underline this aspect). Moreover, the discrepancy between neutralizing and binding activity in this context suggests for the first time a potential role of affinity maturation as an important contributor to vaccination efficacy, as indicated in the discussion.

Reviewer 1 (as well as reviewer2) also rightfully indicates poor (or even absent) explanations in the figure legends. We are grateful for the indications and we apologize for the poor text and for the copy /paste mistakes which involved the figures. In particular the legend of figure 2 was a provisional text pasted instead of the correct one and panel b of figure1 was a duplicate of panel d (which of course caused the impossibility to fully understand the figure. Therefore all  the legends (and parts of the figures and captions) were extensively rewritten, and the paste errors amended.

Yellow underscoring in the revised version of the manuscript indicate parts of the text modified according to the reviewers comments. In addition, green underscoring indicates text modified to improve english style, as well as clarity and completeness of some concepts.

Reviewer 2 Report

The present work describes the cross-reactivity of anti-spike SARS-CoV-2 antibodies from different vaccinated and infected groups over time. The study is interesting to determine differences among vaccinated and/or infected groups, and also to establish more accurate vaccination programs. There are some points to be clarified:

  • Line 73. Please give a reference number.
  • Line 88. Put the microliter symbol properly.
  • Lines 50, 57. Double parenthesis, please correct.
  • Line 125. The name SARS.CoV2 is not correct.
  • Lines 172, 175. Please type correctly the name of lineage B.1.1.7.  
  • Figure 1 footnote. Authors should lines 201 to 208 belong to this figure. Symbols must be defined in the footnote. Please specify the dataset in (b) and (d).
  • Figure 2 should be better explained in text and footnote.
  • Figure 3 footnote and text should be better explained. Symbols must be defined in panels.
  • Figure 4. The number of individuals is very scarce, especially for infected and further vaccinated individuals. You should highlight this fact in the text. Moreover, there is no statistical analysis of these data.
  • Discussion. Lines 20-23 are unsubstantial and must be deleted. 
  • Line 40, please specify the reference numbers for "other studies".

Author Response

Reviewer 2 acknowledges the scientific interest of the manuscript and indicates very precisely a few points that require corrections. We are grateful for the thorough analysis of the manuscript, which will greatly improve it: we have amended (in yellow) each one of the indicated mistakes/corrections . In particular we apologize for not having deleted the first lines at the beginning of the discussion, which was a leftover from the template provided by the journal.

Yellow underscoring in the revised version of the manuscript indicates the parts of the text modified according to the reviewers comments. In addition, green underscoring indicates text modified to improve english style, as well as clarity and completeness of some concepts.

Round 2

Reviewer 1 Report

authors have appropriately addressed concerns raised by this reviewer.

Reviewer 2 Report

Dear authors,

Thank you very much for improving the manuscript.

Kind regards